# A Gradient-Based Self-Organizing Neural Network

## Abstract

Designing the architecture of deep neural networks (DNNs) is a cumbersome task that requires significant human expertise. There have been attempts to automatize this task, mainly by considering DNN architecture parameters such as the number of layers, the number of neurons and the activation function of each layer as hyper-parameters, and using an external method for optimizing them. This framework is computationally too expensive since each hyper-parameter optimization iteration required a whole model training. Here we propose Farfalle Neural Networks, a novel recurrent structure in which neurons connect each other through their trainable embeddings. Hence, important architecture features such as the number of neurons in each layer and the wiring among the neurons are automatically learned during the training process. We theoretically prove that the proposed model can replace a stack of dense layers, which is used as a part of many DNN architectures. By comparing the performance of dense networks and FNNs we show FNNs achieve comparable or higher accuracy using significantly fewer parameters.

## 1 Introduction

During the last few years, deep neural networks (DNNs) have played an incomparable role in addressing challenging problems of different areas, most impressively computer vision (Simonyan & Zisserman (2014); He et al. (2016)). However, well-designing the architecture of a DNN requires a lot of expertise and is usually a cumbersome and time-consuming task, since many possible configurations are assessed before the best one is found. Additionally, the performance of different architectures heavily relies on the nature of the task and the data.

There are several studies (Zagoruyko & Komodakis (2016); Szegedy et al. (2016)) providing insight on how to tune models with state of the art performance for different tasks. Moreover, many studies (Zoph & Le (2016); Cai et al. (2018); Rohekar et al. (2018); Zhong et al. (2018)) have been accomplished recently to avoid architecture engineering. These methods are based on network architecture search over the space of possible architectures (Elsken et al. (2019)). However, these approaches control the structure externally, hence adding a computational overhead when training the model. On the other hand, some studies (Li et al. (2016); Han et al. (2015)) show that it is possible to reduce the number of network parameters and still achieve comparable performance. These smaller networks have the benefit of requiring less storage space and less computational power during inference. These findings also show that there is still much room for improvement in designing efficient neural network architectures.

In this paper, we propose a new neural network model, called Farfalle Neural Network (FNN) in which the trainable parameters are not the weights on the connections between neurons. Instead, the network learns embedding vectors for neurons and uses these vectors to determine the weights of neural connections. More importantly, in the proposed method, instead of hand-crafting the network architecture, it is learned during the training process. The connections between neurons are indirectly specified according to neuron embeddings. Therefore, the proposed network configures its structure itself during the training process given solely the number of nodes and an upper bound on the network depth. We also establish the effectiveness of our models through various experiments. In particular, we show that our model is able to replace fully connected networks achieving higher performance with a 70% reduction in the number of parameters.

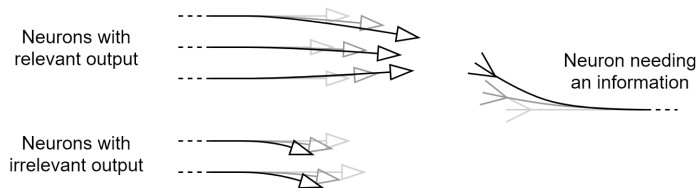

Figure 1: The intuition behind floating neurons. The connections between floating neurons are defined based on their relative similarities. Thus, neurons needing an information and providers of it tend to approach each other.

## 2 RELATED WORK

Recent proposed methods for automatic design of neural networks are commonly focused on treating the architecture decisions as hyper-parameters. These methods use either a supervised or an unsupervised approach to optimize these hyper-parameters. There are several approaches (Zoph & Le (2016); Baker et al. (2016); Zhong et al. (2018)) that utilize reinforcement learning for effectively searching the design space. These approaches usually have a lot of computational overhead because of their need to compute the model's accuracy during the search in the space of architectures Elsken et al. (2019). To circumvent this issue, Smithson et al. (2016) uses another neural network to estimate the trained model's accuracy. However, training the estimator network is itself a computationally expensive task. Rohekar et al. (2018) propose a lightweight unsupervised approach using Bayesian network structure learning. Using this approach, they replace fully connected layers at the end of known networks such as VGGNet with smaller models while still showing a comparable performance in accuracy. Note that while the reported results show the effectiveness of this method, it still optimizes the structure externally. Hence, it requires an additional environment setup and have an external overhead in the learning process, though the latter is reported to be reasonably small.

One of the key ideas in our proposed method is to assign embedding vectors to neurons of the network and use the attention mechanism to relate them. Similar ideas appear in the Transformer network (Vaswani et al. (2017)) and CapsNet (Sabour et al. (2017)). In Transformer networks (Vaswani et al. (2017)) input words and their positional information are embedded in a low-dimensional space. However, they utilized a specific case of attention called self-attention to relate different parts of the sequence. In addition, the embeddings used in that architecture are not trained for the purpose of structure learning. CapsNet (Sabour et al. (2017)) considers an output vector for each capsule and routes the outputs from one capsule to the next layer's capsules according to its ability to predict the output vector of those capsules. However, CapsNets still use weight matrices between neurons and also are not able to self configure their structure.

## 3 FARFALLE NEURAL NETWORKS

In traditional neural networks, the number of neurons in each layer and the arrangement of the neurons is fixed. This rigid configuration prevents straightforward optimization of the network structure during the training process. Therefore, finding a network with proper structure requires testing a lot of configurations.

In contrast, FNNs utilize a new type of neurons which can float and find the most suitable neurons to obtain information from them. The connections between these *floating neurons* are defined based on their relative similarities. Thus, during the training process, relevant neurons move toward each other to strengthen their connection. Figure 1 shows how these neurons float to obtain more relevant information.

### 3.1 FLOATING NEURONS

A floating neuron gathers information from relevant neurons at its input, transforms it with a trainable transformation, and emits the result at its output. In order to avoid confusion, we might refer to a neuron's input as its head. Similarly, we sometimes refer to its output as the neuron's tail. Inputs

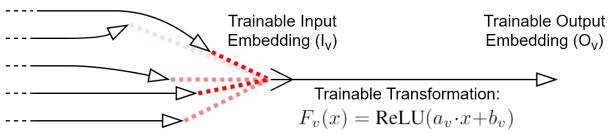

Figure 2: A schematic representation of a floating neuron and its connections. Each neuron has an input embedding, an output embedding, and a transformation function.

and outputs of these neurons are embedded in a $d$-dimensional space. These embedding vectors regulate the weights connecting relevant neurons and are updated during the training process. Figure 2 shows a schematic representation of a floating neuron and its connections. Specifically, a floating neuron $v$ consists of three parts:

- **Input embedding:** A trainable $d$-dimensional vector $I_v$ which indicates the coordinates of the neuron's head. The similarity between this vector and output embedding of other neurons determines the connection weights between this neuron and other neurons using the attention mechanism.

- **Transformation function:** A trainable nonlinear function $F_v$ that transforms the gathered information. The transformation used in this study is of the form $F_v(x) = \text{ReLU}(a_v \cdot x + b_v)$ where $a_v$ and $b_v$ are neuron-specific trainable parameters.

- **Output embedding:** A trainable $d$-dimensional vector $O_v$ which indicates the coordinates of the neuron's tail.

In addition to normal floating neurons, there are two custom types of floating neurons: Input neurons, which receive the input of the whole network and output neurons, which provide the processed data to the outside. Consequently, input neurons do not have input embedding and output neurons do not have output embedding.

## 3.2 Construction of Multi-Layer Floating Neural Networks

Before introducing FNNs, we discuss how to employ floating neurons in a layered structure. In order to form such a network, floating neurons are grouped in layers. The neurons at each layer obtain their values from neurons at the previous layer. The connections between neurons of two consecutive layers are defined based on the attention mechanism.

Formally, suppose $v$ is a neuron in the layer $i + 1$ and $u_1, u_2, \ldots, u_M$ are neurons of the previous layer, i.e. layer $i$. The weights connecting $v$ to related neurons is defined by

$$w_1, w_2, \ldots, w_M = \mathcal{N}(I_v^T O_{u_1}, I_v^T O_{u_2}, \ldots, I_v^T O_{u_M}) \tag{1}$$

where $I_v$ is the input embedding of neuron $v$, $O_{u_i}$ is the output embedding of neuron $u_i$, and $\mathcal{N}$ is a normalization function. For normalization, one can choose softmax function to force each neuron's input to be a convex combination of the outputs of the neurons in the previous layer. We found $l2$ normalization to work best in our experiments and thus the following function is used for normalization

$$\mathcal{N}_{l2}(x_1, x_2, \ldots, x_n) = \frac{x_1}{\sum_i x_i^2}, \frac{x_2}{\sum_i x_i^2}, \ldots, \frac{x_n}{\sum_i x_i^2}. \tag{2}$$

Utilizing these weights, given $y_1, y_2, \ldots, y_M$ as the values of neurons $u_1, u_2, \ldots, u_M$, the output value of neuron $v$ will be $F_v(\sum_m w_i y_i)$ where $F_v$ is the transformation function of neuron $v$. To efficiently connect neurons $v_1, v_2, \ldots, v_N$ of the layer $i + 1$ to neurons $u_1, u_2, \ldots, u_M$ of the layer $i$, let $I = [I_{v_1} | I_{v_2} | \cdots | I_{v_N}]$ be the concatenation of input embedding vectors of the layer $i + 1$ and $O = [O_{u_1} | O_{u_2} | \ldots | O_{u_M}]$ be the concatenation of output embedding vectors of the layer $i$. Then, the weights connecting neurons of these two layers are determined by

$$W = \widetilde{\mathcal{N}}\left(I^T O\right) \tag{3}$$

where $\widetilde{\mathcal{N}}$ is a function normalizing each row of its input matrix according to the normalization function $\mathcal{N}$. Finally, the output of layer $i + 1$ will be

$$Z = \widetilde{\mathcal{F}}(WY) \tag{4}$$

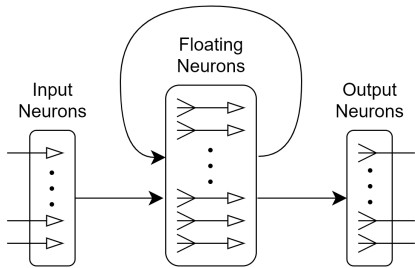

Figure 3: The architecture of Farfalle Neural Networks.

where $Y$ is the vector containing output values of layer $i$ and $\widetilde{\mathcal{F}}_i = F_{v_i}$.

Using this model, the number of parameters required to connect a layer of size $M$ to a layer of size $N$ is of the order $O(d(M + N))$. In contrast, in dense models the number of parameters for connecting two such layers would be of the order $O(MN)$. Hence, by using this model one of the limitations in designing neural networks, that is the huge number of parameters in the weight matrices, is resolved.

### 3.3 CONSTRUCTION OF FARFALLE NEURAL NETWORKS

An FNN is a recurrent network of floating neurons. Hence neurons in this architecture are used in iterations. In each iteration, floating neurons receive the output of the previous iteration along with the input. Figure 3 represents the architecture of FNNs. The expanded architecture is available in Supplementary Figure 7. This recurrent structure allows neurons to process high-level information along with low-level features. Also, the floating nature of these neurons allows the network to balance the number of neurons employed in different levels of abstraction. Supplementary Figure 8 illustrates how this recurrent structure allows the network to combine information from different levels of abstraction. This flexibility allows the architecture to evolve during the training.

Before describing the data flow of FNNs, let's define an auxiliary function. For two groups of neurons $V$ and $U$, equation (3) defines the weights connecting neurons in $V$ to neurons in $U$. In this equation $O$ and $I$ are the concatenation of output embeddings of neurons in $U$ and the concatenation of input embeddings of neurons in $V$ respectively. Given $Y$ as the values of neurons in $U$, equation (4) describes the output values of neurons in $V$. Combining these equations results in

$$\text{Step}(V, U, Y) = \widetilde{\mathcal{F}}(\widetilde{\mathcal{N}}\left(I^T O\right) Y)\tag{5}$$

where $\widetilde{\mathcal{N}}$ is a function normalizing each row of its input matrix and $\widetilde{\mathcal{F}}$ transforms values of each floating neuron using its own transformation function. The function Step can be seen as the combination of attention and neuronal transformation.

Constructing an FNN for the input size of $R$ and output size of $S$ needs $R$ input floating neurons for feeding data to the network, $N$ floating neurons for processing data in $k$ iterations, and $S$ output floating neurons for the final deduction from hidden neurons. Here, $N$ and $k$ are the hyperparameters of the network. Let's call the input neurons $\mathcal{I} = i_1, i_2, \ldots, i_R$, the hidden neurons $\mathcal{V} = v_1, v_2, \ldots, v_N$, and the output neurons $\mathcal{O} = o_1, o_2, \ldots, o_S$. The following procedure describes the flow of the network:

1. Each input neuron assigns an embedding vector to its input variable after applying its transformation function, i.e. given input $x = (x_1, x_2, \ldots, x_R)$, input neurons will provide values $Y_0 = F_{i_1}(x_1), F_{i_2}(x_2), \ldots, F_{i_R}(x_R)$ at locations $O_{i_1}, O_{i_2}, \ldots, O_{i_R}$.

2. In the first step of the iterative part, hidden neurons process the data provided by input neurons. Thus, the resulting values of this step is $Y_1 = \text{Step}(\mathcal{V}, \mathcal{I}, Y_0)$.

3. In iteration $1 < j \leq k$, each hidden neuron process $Y_0$ along with the outputs of all hidden neurons in the previous iteration $(Y_{j-1})$ and produce $Y_j$. Indeed, given $Y_0$ and $Y_{j-1}$ hidden floating neurons will provide $Y_j = \text{Step}(\mathcal{V}, [\mathcal{I}|\mathcal{V}], [Y_0, Y_{j-1}])$.

4. Finally, utilizing output neurons the final output will be $\text{Step}(\mathcal{O}, \mathcal{V}, Y_k)$.

The following theorem shows how an FNN with its recurrent structure can model a multi-layer floating neural network.

**Theorem 1.** *Every multi-layer floating neural network with $l$ layers, total of $N$ floating neurons, and embedding dimension $d$ can be modeled by an FNN containing $N + 1$ floating neurons with embedding dimension of $l \cdot d$ which iterates for $l$ iterations.*

*Proof.* Suppose a multi-layer floating neural network $N1$ with layer sizes of $n_1, n_2, \ldots, n_l$ is given. Let $i_1, i_2, \ldots, i_R$ be the input floating neurons, $v_1^j, v_2^j, \ldots, v_{n_j}^j$ be the floating neurons in layer $j$, and $o_1, o_2, \ldots, o_S$ be the output floating neurons of the network.

To construct an FNN $N2$ with the same functionality, define a network with input neurons $i_1', i_2', \ldots, i_R'$, output neurons $o_1', o_2', \ldots, o_S'$, and hidden neurons $v_0', v_1'^1, v_2'^1, \ldots, v_{n_l}'^l$ such that $v_i'^j$ correspond to hidden neuron $v_i^j$ of $N1$. Let all floating neurons in $N2$ have the same transformation function as their corresponding neuron in $N1$ and $v_0'$ be a neuron with $F_{v_0'} = 0 \cdot x$ and $I_{v_0'} = O_{v_0'} = (\epsilon, \ldots, \epsilon)$ where $\epsilon$ is a negligible values. Before defining neural embeddings of $N2$, for $0 < j \leq l$ define

$$E_j(x_1, x_2, \ldots, x_d) = (\overbrace{0, \ldots, 0}^{(j-1) \cdot d}, x_1, x_2, \ldots, x_d, \overbrace{0, \ldots, 0}^{(l-j) \cdot d}).$$

This implies that

$$E_i(x)^T E_j(y) = \begin{cases} x^T y & i = j \\ 0 & i \neq j \end{cases}. \tag{6}$$

Now, define all remaining embedding vectors of $N2$ by

$$O_{i_r'} = E_1(O_{i_r}) \qquad\qquad I_{o_s'} = E_1(I_{o_s})$$
$$O_{v_i'^j} = E_{(j \bmod l)+1}\left(O_{v_i^j}\right) \qquad\qquad I_{v_i'^j} = E_j\left(I_{v_i^j}\right).$$

These definitions along Eq. (6) result that in the first iteration, the only neurons which have a non-zero connection with input neurons are $v_1'^1, v_2'^1, \ldots, v_{n_1}'^1$. In addition, the weights of these connections are the same as the connections in the first layer of $N1$. Similarly, by mathematical induction over $j$ we can state that floating neurons $v_1'^j, v_2'^j, \ldots, v_{n_j}'^j$ can only connect to $v_1'^{j-1}, v_2'^{j-1}, \ldots, v_{n_{j-1}}'^{j-1}$, the connection weights are the same as those in layer $j$ of $N1$, and iteration $j$ is the first time $v_1'^j, v_2'^j, \ldots, v_{n_j}'^j$ can get nonzero values. This implies that the connections of hidden floating neurons of $N2$ in $l$ iterations, form a chain similar to the architecture of $N1$. Finally, $I_{o_s'}$ is just connected to $v_1'^l, v_2'^l, \ldots, v_{n_l}'^l$ with the same weights as in $N1$. This completes the proof. It is worth mentioning that neuron $v_0'$ is defined to eliminate the division by zero occurred in Eq. (2) when a neuron does not have a non-zero dot product to any input neuron. $\square$

Note that by choosing a right value for $d$, traditional neural networks can be replaced with multi-layer floating neural networks introduced in this paper. Additionally, Theorem 1 implies that the farfalle neural networks are more general than multi-layer floating neural networks. Thus, as a corollary we note that it is possible to replace traditional multi-layer neural networks with the recurrent structure of FNNs.

Furthermore, there is a repeated observation that replacing weights of a neural network with their low-rank approximations gives a comparable (or even improved) performance (Sainath et al. (2013); Denil et al. (2013); Denton et al. (2014)). Such approximation allows us to replace dense networks with FFNs with small values of $d$ and thus reasonable number of parameters.

Utilizing this model, more flexibility in specifying the layer sizes is provided accordingly. Furthermore, since we do not need to have weight parameters explicitly, we can consider the whole network as a fully connected structure in which each (hidden) neuron can be potentially connected to all other neurons and even itself.

### 3.4 SCALABILITY & PRODUCTION

During training, the weight matrix $W$ needs to be computed to apply the normalization function. Since the number of elements in this matrix is quadratic in the number of neurons, it is possible that this matrix becomes quite big. However, after the training, it is possible to workaround the normalization step by updating the matrix $I$. Specifically, using the same notation as the last section, it is enough to replace $I_v$ with $\frac{I_v}{\sum_i (I_v^T O_{u_i})^2}$. This simplification significantly reduces the required memory space during inference since the dimension of the embedding space is usually much smaller than the number of neurons.

Furthermore, although the need to compute the weight matrix during the training imposes a practical limit on the maximum number of neurons in FNN, the upper bound is still very large. Additionally, it is possible to stack FNNs similar to normal layers to employ more floating neurons. Such structure does not allow the use of information from all deeper layers but is still much more flexible than commonly used dense layers.

## 4 RESULTS & DISCUSSION

In the following subsections, we present comparison results of our model with other DNN architectures on CIFAR (Krizhevsky et al. (2009)), a widely-used image classification dataset. First, we compare FNNs with a dense model and show that our model can outperform them. Then we discuss some characteristics of FNNs, such as their ability to learn locality. In the final section, we discuss how our model can be integrated with existing convolutional neural networks.

### 4.1 COMPARISON WITH DENSE MODELS

Although dense models are not among the state of the art models for neither datasets, the goal of this section is to establish the effectiveness of our model in comparison with dense models. We do not claim that our model, in any way, can directly outperform highly specialized models such as convolutional neural networks. Instead, we demonstrate how our model may be used in conjunction with CNNs in subsequent sections.

In order to obtain a proper baseline for CIFAR, we used HyperOpt Bergstra et al. (2013) to search among fully connected layers with up to 4 hidden layers. Number of neurons in each layer was chosen from $\{500, 1000, 1500, 2000\}$. The top performing model was a fully connected network with 2 hidden layers of 2000 and 500 neurons, in order from input to output.

We trained a multi-layer floating network with the same structure and embedding size 256. We also trained a FNN the same number of neurons and embedding size. The number of iterations was set to 2. Furthermore, we trained a smaller fully connected network with the same number of parameters as our models by reducing the number of neurons in the first hidden layer of baseline to 850. We trained all four models using Adam (Kingma & Ba (2015)) algorithm for 200 epochs. In all models we applied 0.1 dropout (Srivastava et al. (2014)) rate of the input and used ReLU (Glorot et al. (2011)) for activation function.

Summarized results are shown in Table 1. It is evident that our models significantly outperform the fully connected networks. This is especially significant since there is more than 70% reduction in number of parameters in multi-layer floating network with respect to the fully connected network of the same architecture. The performance gap is worsened when comparing with the smaller base-line.

The results show that FNNs are able to obtain comparable results. The slightly better performance of multi-layer floating network perform with respect to FNN is not unexpected. The structure for the multi-layer floating network is the same as the baseline which was chosen by searching through the architecture space. Moreover, the FNN was trained with almost no hyper-parameter tuning.

### 4.2 ANALYSIS OF LEARNED EMBEDDINGS

We analyzed the embeddings of an FNN with 1024 neurons, 256 embedding size, and 1 iteration trained on MNIST (LeCun et al. (1998a)). The training setting was the same as the previous section.

| Model | Top-1 Accuracy(%) | # of Parameters |
|---|---|---|
| Baseline | 53.69 | 7151510 |
| Small-Baseline | 52.29 | 2077850 |
| Floating Network | **63.18** | 2080156 |
| FNN | 63.01 | 2080156 |

Table 1: The performance of FFNs in comparison with baseline method.

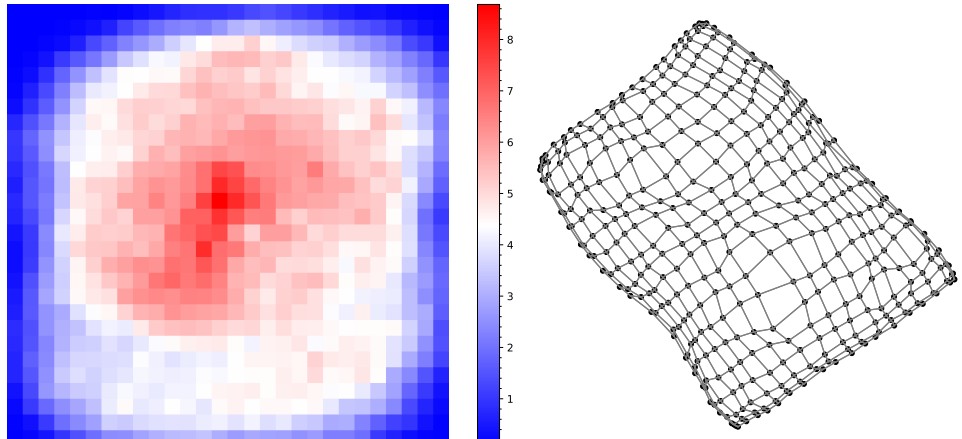

Figure 4: The $l2$-norm of each input neuron's embedding is calculated and plotted at its corresponding cell.

Figure 5: The input embeddings are projected to 2D space using UMAP. Only neurons in the inner $20 \times 20$ box are included. Neurons corresponding to adjacent cells are connected with a line.

In Figure 4, each cell is colored according to the $l2$-norm of its corresponding neuron's output embeddings. It can be seen that marginal neurons have much lower $l2$-norms. This means that the embeddings are much smaller for these neurons, and so they have little effect on the model's output. Note that this is expected since the marginal pixels in MNIST images seldom provide useful information.

Figure 5 depicts the projection of the learned embeddings to 2D space. The projection is performed using Uniform Manifold Approximation and Projection (UMAP, McInnes et al. (2018)). Neurons corresponding to adjacent cells are connected with a line. Marginal neurons are excluded in order to obtain a better projection of the embeddings. It is apparent in the figure that the learned embeddings respect the locality of pixels, so a pair of pixels close to each other have similar embeddings. Hence, it can be seen that FNN is able to assign meaningful embeddings to the neurons.

### 4.3 INTEGRATION WITH CNNS

Convolutional neural networks (LeCun et al. (1998b)) are widely used in image classification tasks and have been able to produce state of the art results. Commonly in such networks, convolutional layers are employed for feature extraction. The extracted features are then fed into several fully connected layers for classification. We propose that FNNs can be used to replace these fully connected layers.

To test this, we compared FNN with a baseline model on CIFAR10 and CIFAR100 datasets. We use a slightly modified version of VGG16 Simonyan & Zisserman (2014) which is suitable for CIFAR as our baseline. In this version, all layers after the last max-pooling layer are replaced with fully connected network with one hidden layer of 512 neurons. Batch normalization and a dropout rate of 0.5 is applied after the hidden layer. The same dropout rate is also applied before the hidden layer. We used ReLU as the activation function. A similar model has been used as a baseline in other studies (Rohekar et al. (2018); Li et al. (2016)).

| Model | Top-1 Accuracy(%) | |
| --- | --- | --- |
| | CIFAR10 | CIFAR100 |
| VGG16 + Dense | 92.98 | 73.19 |
| VGG16 + FNN | **93.51** | **73.33** |

Table 2: Top-1 accuracies when using normal dense layers or a FNN in VGG16.

We compare this baseline with an alternative architecture consisting of an FNN with 1024 neurons after the last max-pooling layer. The number of iterations was set to 3. We used Stochastic Gradient Descent (SGD) with 0.9 momentum Rumelhart et al. (1988) for training and employed 0.0005 weight decay regularization.

The maximum test accuracy of both models are presented in Table 2. It is evident that our model outperforms the dense layers on both datasets.

## 5 FUTURE WORK

We established that FNNs are able to replace and outperform fully connected layers.

FNNs are an extreme case of sharing embeddings between layers of a multi-layer floating neural network. We note that it is possible to use different patterns of sharing. Exploring this area might specially be interesting in the case of auto-encoders.

We conjecture that some of the neuron embeddings trained to solve one task can be used to solve similar tasks. The input neuron embeddings projection in Figure 5 supports this conjecture. However, this task is not trivial, and we leave it as a future work.

Another interesting direction is to assign different activation functions to different neurons in an FNN. This feature allows a combination of activation functions to be used together which is, to the best of our knowledge, something less explored in hand-crafted architectures.

## 6 CONCLUSION

In this paper, we introduced a method to learn the network structure internally during training. This was done mainly based on the new approach of assigning parameters to the neurons instead of the connection between them. Using this approach, we introduced a novel neural network structure called Farfalle Neural Network. We established through experiments that this new structure can outperform dense layers in various scenarios while even sometimes using significantly (70%) lower number of parameters. We also discussed how this approach could significantly reduce the memory requirements during the inference process.

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

# A    ABLATION STUDY

In this section, we investigate the effect of three hyper-parameters, specifically number of iterations, embedding size, and normalization function, on the performance of FNNs. To assess the effect of each hyper-parameter, we set all other hyper-parameters to a default value and evaluate model performance for various values of this hyper-parameter. The default values are 3 for the number of iterations, 256 for the embedding size, and $l2$ for the normalization function. In all experiments, the FNN consists of 1024 neurons. The results are presented in Figure 6.

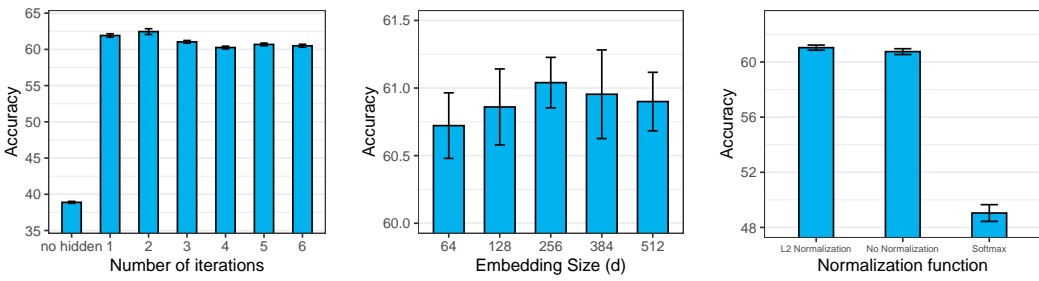

Figure 6: The effect of different parameters on the performance of FNNs. The default values for number of iterations, embedding size, number of neurons, and normalization function are 3, 256, 1024, and l2 respectively. The default values are used if not specified.

# B    INTEGRATION WITH OTHER CNNS

In this section we present results of using FNNs instead of the fully connected layers at the tail of other convolutional networks, specifically Residual Networks He et al. (2016). We compare performance of our models with baseline models on CIFAR100 dataset.

We train ResNet18 as our baseline and compare it with a variant of ResNet18 with an FNN instead of the tail fully connected layer. The FNN has 512 neurons with 64 embedding size. The number of iterations is set to 1 so the network is equivalent with a multi-layer floating network with one hidden layer. For this task, we fixed the transformation function of the input neurons to identity. The results are presented in Table 3.

| Model | Top-1 Accuracy(%) |
|---|---|
| ResNet18 + Dense | 76.55 |
| ResNet18 + FNN | 76.60 |

Table 3: The performance of ResNet18 and a variant of it with an FNN instead of the tail fully connected layer on CIFAR100.

SUPPLEMENTARY FIGURES

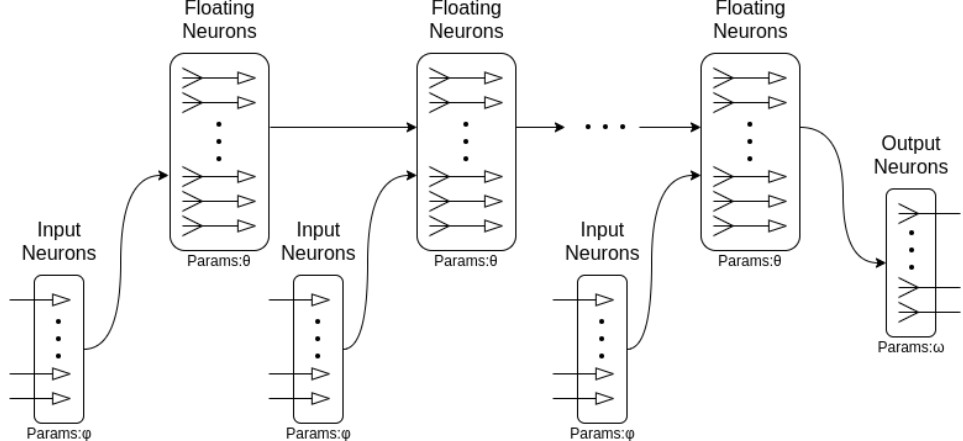

Figure 7: Unrolling Farfalle Neural Networks. FNNs have a recurrent block of floating neurons. Unrolling this structure results in a layered structure of floating neurons, in which the parameters are shared between layers. In addition, input is fed into each hidden layer.

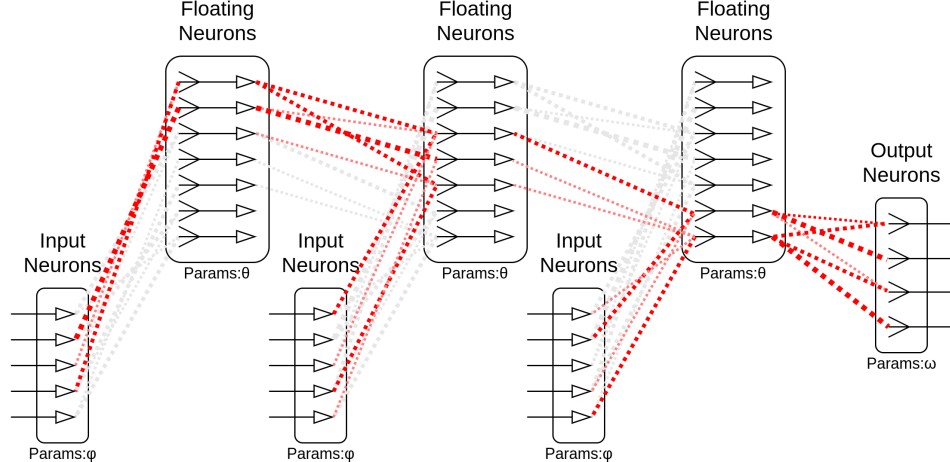

Figure 8: The recurrent structure allows the network to balance the number of neurons employed in different levels of abstraction. This illustration employs unrolled structure to show how neurons can combine information from different levels of abstraction. Red lines specify active connections (connections that significantly influence output) of each iteration.

