# OpenReview forum: "A Gradient-Based Approach to Neural Networks Structure Learning"
_ICLR.cc/2020/Conference — Reject_

### Official Review · AnonReviewer3 · 2019-10-23
**Official Blind Review #3**

**Rating:** 6

**Review:**

This paper introduces a new neural network architecture, in which all neurons (called "floating neurons") are essentially endowed with "input" and "output" embedding vectors, the product of which defines the weight of the connection between any two neurons. The authors discuss two network architectures employing floating neurons: (a) multi-layer floating neural networks and (b) farfalle neural network (FNN), in which there is one hidden layer, but additional recurrent connections are introduced between the hidden neurons.

As mentioned by the authors, the proposed architecture is similar to architectures employing low-rank weight matrix factorization. In my opinion, the main novelty lies in: (a) "floating neuron" interpretation, (b) additional weight matrix normalization, and (c) FNN architecture similar to that of a "floating neuron" RNN network with additional restrictions.

I find the proposed idea to be promising and quite intriguing, but I think that the paper has some room for improvement and provided empirical evidence might be insufficient (including for understanding the importance of individual model components), which in turn makes the claims of potential practical attractiveness less justified. I will be happy to update the final score provided with more compelling arguments or empirical analysis of the proposed architecture.

Addressing the following issues might greatly improve the quality of the paper:

1. In Section 4.1, the authors compare FNN and DNN on MNIST and CIFAR10 datasets. My concern is that the authors pick a seemingly arbitrary DNN architecture (just a single one) and restrict comparison to it. One issue is that ~50% accuracy on CIFAR10 can be easily demonstrated by a variety of 5-layer DNN architectures including those much smaller, with just ~600k parameters (!) and possibly even lower. This makes the 90% parameter reduction claim not particularly meaningful. And why were models matched based on the total number of neurons, but not, say, the total number of parameters, or other measures? I believe that these questions require additional discussion and empirical evidence. Just as an example, if it was possible to sample (potentially randomly) different DNN architectures (with a reasonable parameter prior) and compare them with FNNs on a 2D accuracy-parameters plot (or using other important metrics), it would provide much more information to the reader.

2. Another important point that I would like to make is that there is much more that can be done to explore the hyper-parameter space of FNN to isolate which particular factors play a decisive role in its superior accuracy. The authors present us with a specific choice of the normalization function, and values of k and d, but it would be very informative to study how results change when different choices are considered. FNNs differ from DNNs in at least three aspects: usage of low-rank factorization, weight normalization and recurrent structure. How important are these individual aspects? Are some of them redundant, or almost redundant, or do FNNs require all of these components to achieve their peak performance? In other words, I believe that a careful ablation study would greatly improve this publication.

3. As a minor note, I think that the statement that FNNs "are more general" than floating neural networks is only partially correct. If I am not mistaken, FNN can also be "unrolled" and represented as a multi-layer floating neural network with additional parameter sharing. Also, the computational complexity of the constructed FNN (in Theorem 1) appears to be significantly higher than that of the floating neural network (especially for high l). This would imply that FNNs do not necessarily supersede multi-layer floating neural networks, at least when the computational complexity is of importance.

4. There are a few minor misprints throughout the text. For example, in "0<j<=j" in the proof of Theorem 1, or in "R output floating neurons for the final deduction from hidden neuron" (output should be S). Also, I could not find information about the value of d used in the described experiments (which I estimated to be 256; is this correct?).

5. In Section 4.3, the authors propose to use FNNs for the final layers of conventional CNN architectures. The issue is that the VGG16 network chosen for experiments was probably picked because it uses several large fully-connected (FC) layers in its tail whereas all more recent and efficient CNN architectures actually gravitate towards a smaller single FC layer. It is possible that FNNs could still be used in FC layers of these modern networks as well (especially with a large number of classes). But additional empirical results for these architectures would, in my opinion, be much more convincing.

Updated: The authors updated the text and addressed many of my questions. In my opinion, this improved the paper and made some of its claims much better justified. I change the rating to "Weak Accept".

**Experience Assessment:**

I have published one or two papers in this area.

**Review Assessment: Checking Correctness Of Derivations And Theory:**

I assessed the sensibility of the derivations and theory.

**Review Assessment: Checking Correctness Of Experiments:**

I assessed the sensibility of the experiments.

**Review Assessment: Thoroughness In Paper Reading:**

I read the paper at least twice and used my best judgement in assessing the paper.

---

> ### Author Response · Authors · 2019-11-14
> **Response to review**
>
> Thank you very much for reviewing our paper and proposing valuable comments.
>
> We have responded to your comments below.
>
> 1.
> 	(I) To alleviate your concern regarding the baseline, we used HyperOpt to search fully connected architectures with up to 4 hidden layers. In the new revision of our paper, we replaced our baseline with the top-performing model found in the search. We also provided results for a smaller baseline with the same number of parameters as in our models.
> 	(II) The rationale behind matching networks based on the number of neurons was converting a fully connected network to an FNN in two steps. Initially, the fully connected networks are converted to a multi-layer floating network. The reduction in the number of parameters occurs in this step. The multi-layer floating network is then replaced with an FNN to allow self-configuration. However, we understand why the initial results would not be convincing. In the new revision, we added the performance results of a multi-layer floating network with the same structure as the baseline. Additionally, we reduced our claim of parameter reduction according to the new results since we now understand that our previous comparison might have been unfair.
>
> 2. As you suggested, we have added an ablation study in Appendix A, showing how each parameter affects the model's performance. Specifically, the results show that all of the three aspects mentioned are important to obtain peak performance.
>
> 3. The constructed FNN in Theorem 1 is provided merely to establish an upper-bound, and its only purpose is to prove the ability of FNNs to replace the multi-layer version. This ability is why we referred to them as more general. The FNNs are superior because they do not require an assignment of neurons to layers. As you mentioned, the superiority stems from the additional parameter sharing. Also, even though we do not theoretically prove that it is possible to do this efficiently, we show that it is possible to use FNNs in practice with a reasonable embedding space dimension.
>
> 4. We apologize for the mentioned misprints. We have corrected such mistakes in the new revision.
>
> 5. You are correct that part of our reason for choosing VGG is because it uses a larger FC layer at the end. We have added the results for using FNN instead of the FC layer at the tail of ResNet18 on CIFAR100 in Appendix B. However, in ResNet, most of the work is being done with CNNs, and the FC layer is minimal. As a result, the improvement when using our model is limited. We agree that the improvement might be increased when running on datasets with an even larger number of classes, such as ImageNet. However, training a model on ImageNet in such limited time requires a lot of computational resources, which, unfortunately, we do not have.
> We would like to emphasize that our goal in this paper is to propose a self-configuring structure of neurons which can replace FC networks. It is not our intention to improve existing models in a specific field, such as computer vision. We used CIFAR and MNIST merely because they are well-known. Furthermore, we experiment with CNNs only to show the integrability of our model with existing layers.

---

### Official Review · AnonReviewer1 · 2019-10-23
**Official Blind Review #1**

**Rating:** 3

**Review:**

Overall the paper is easy to read and I welcome that.

I like the idea of using node-level embedding instead of pairwise weights to learn a low-rank weight representation. However, I am more skeptical about using this in a recurrent architecture and claiming that this is structure learning. The empirical results do not provide sufficient evidence that this performs structure learning.

1. Theorem 1 seems rather straightforward because the FNN has much more representational power in the sense that its number of parameters is O(Nld) whereas the multi-layer version has O(Nd/l) parameters (in the uniform-width case).

 -- A more interesting question when it comes to structure learning is this: Suppose the best architecture for task A is shallow-and-wide while for task B is deep-and-narrow, each requiring roughly the same number of parameters. Can I use the proposed FNN with a similar number of parameters to learn the corresponding architecture for A and B respectively, without the need to figure out which is which? There is no evidence, analytical nor empirical, in this work, that suggests that this is the case.

2. Section 4. It would be interesting to try baselines that have roughly the same number of parameters as the proposed FNN. Also, the choice of d (embedding size) and the number of iterations can be viewed as making architectural decisions. How were they chosen? Assuming that the same amount of computational resource is spent on searching through baseline architectures as well, could the results have been different from those in Table 1?

There are interesting ideas in this work but in its present form I cannot yet recommend acceptance.


**Experience Assessment:**

I have read many papers in this area.

**Review Assessment: Checking Correctness Of Derivations And Theory:**

I carefully checked the derivations and theory.

**Review Assessment: Checking Correctness Of Experiments:**

I carefully checked the experiments.

**Review Assessment: Thoroughness In Paper Reading:**

I read the paper thoroughly.

---

> ### Author Response · Authors · 2019-11-15
> **Response to review**
>
> Thank you very much for reviewing our paper and proposing valuable comments.
>
> We had called our approach structure learning  to emphasize the fact that our model does not need a hand-crafted structure. Still, to avoid any misunderstanding we have updated the title in the new revision of our paper.
>
> We have addressed your comments below.
>
> 1. Your point about the representational power of constructed FNN in Theorem 1 is correct (except for a minor issue that the number of parameters in the multi-layer version is O(Nd) not O(Nd/l)).
> 	(I) However, note that this is merely an upper-bound. The main purpose of this theorem is to prove the ability of FNNs to replace the multi-layer version.
> 	(II) Additionally, we show that it is possible to use FNNs in practice with a reasonable embedding space dimension.
> 	(III) It is worth mentioning that the benefit of using the recurrent version is not a reduction in the number of parameters. Instead, by using the recurrent version, the dimension of (discrete) search space for the number of nodes in different layers reduces from l to 1. This reduction facilitates self-configuration since searching in the continuous space of parameters is much easier than search in the discrete space of such hyper-parameters.
> 	(IV) Regarding your proposal, we would like to first thank you for sharing this idea. Unfortunately, we are not aware of any task where a shallow-and-wide network would perform better than a deep network. Hence we could not find two datasets that satisfy similar conditions as you described. We would appreciate any suggestions you might have in this regard. Furthermore, we would like to clarify that we do not argue that our model can learn a specific structure, such as a deep network, but that it is able to find an internal structure for each task.
>
> 2.
> 	(I) In the new revision of our paper, we replaced our previous baseline with the top-performing model found when searching amongst fully connected networks with up to 4 layers using HyperOpt. We also presented results for a smaller baseline model with approximately the same number of parameters as our FNN, as you suggested.
> 	(II) Note that testing several embedding sizes and tuning the number of iterations does not come computationally even close to a thorough search of possible fully connected architectures (that includes not only optimizing the number of layers but also the number of hidden neurons in each layer). It is worth mentioning that we did not perform task-specific tuning of FNNs to obtain the reported results. So to obtain the latest results, we have spent more computational resources to tune the baseline than we did for our model.
> 	(III) To choose the embedding size, we tested a limited set of embedding sizes (128, 256, 512) on a validation subset sampled from training data of MNIST. The best embedding size (256) was chosen for all the tasks without separate tuning for each task. We reported this value in the revised version of our paper accordingly.
> The number of iterations in the first task (comparing with FC networks) was the same as the number of hidden layers in the fully connected network being compared with our model. When integrating FNNs with VGG we also tested an FNN with 3 iterations and chose it over the FNN with 1 iteration. No additional tuning, such as testing other values for number of iterations, was done.

---

### Official Review · AnonReviewer2 · 2019-10-23
**Official Blind Review #2**

**Rating:** 3

**Review:**

This paper proposes a new architecture based on attention model to replace the fully-connected layers. In this architecture, each neuron is associated with an embedding vector, based on which the attention scores (between two consecutive layers) are calculated, and the computational flow through the layers are derived based on these attention scores. The experiments on MNIST and CIFAR demonstrate some degree of superiority over plan FC layers.

Pros:

1. The idea is indeed interesting and AFAIK, there is no prior works trying to derive embedding for each neuron. The embedding based connection might encourage other follow-up works.

Cons:

1. The writing sometimes seems unnecessarily complicated. For example, the “iteration” in section 3.3 is actually “layer”, right?  I furthermore see no motivation of listing the four items in this section, even the whole section 3.3: they are just re-stating the feedforward process of FNN.
2. I donot believe FC is essential in modern computer vision (CV) tasks, so the better performance over a plain FFN on CV tasks are not that convincing (especially the two datasets are typically regarded as debugging dataset nowadays). I suggest the authors conduct more experiments on Transformer based tasks (e.g., machine translation), since in Transformer, the FFN is quite important. If the replace of FFN using the proposed FNN is successful for Transformer on some large scale task (e.g., WMT14 En-De Translation), this work will be much stronger in terms of empirical performance.

Question:

1. What is the embedding size d in the experiments? If d is large, the complexity comparison in the last paragraph of section 3.2 will not make too much sense.



**Experience Assessment:**

I have published one or two papers in this area.

**Review Assessment: Checking Correctness Of Derivations And Theory:**

I assessed the sensibility of the derivations and theory.

**Review Assessment: Checking Correctness Of Experiments:**

I assessed the sensibility of the experiments.

**Review Assessment: Thoroughness In Paper Reading:**

I read the paper at least twice and used my best judgement in assessing the paper.

---

> ### Author Response · Authors · 2019-11-13
> **Response to review**
>
> Thank you very much for reviewing our paper and proposing valuable comments.
>
> We are delighted that you are interested in our main idea and see the possibility of follow-up works.
>
> However, we think that one of the main contributions of our work is missing from your summary and review. As you pointed out correctly, in Section 3.2, we proposed “floating neural networks”, which assigns embedding vectors to neurons and employs the attention mechanism to replace FC layers. However, please note that in Section 3.3, we proposed “Farfalle neural networks”, which has a recurrent structure. In the recurrent version, data passes through all hidden floating neurons several times, facilitating self-configuration. Furthermore, in Theorem 1, we stated that this recurrent structure can model any FC layer without the need for a search in the vast space of FC architectures.
>
> We have responded to your comments below.
>
> 1. We think there is a misunderstanding regarding the FNN proposed in section 3.3. Section 3.3 is dedicated to describing a recurrent version in contrast with the flat version described in Section 3.2. In the recurrent version, data passes through all neurons several times, whereas, in the flat version, data is passed through each neuron just once. That is why “iteration” is used rather than “layer”. It is worth mentioning that the flat version of our proposed method (introduced in Section 3.2) needs manual assignment of neurons to layers, while the proposed model in Section 3.3 allows self-configuration.
>
> 2. Regarding datasets and empirical performance.
> (I) The goal of this paper is to propose a self-configuring structure of neurons which can replace FC networks. It is not our intention to improve existing models in a specific field, such as computer vision. We used CIFAR and MNIST merely because they are well-known. Furthermore, we experiment with CNNs only to show the integrability of our model with existing layers.
> (II) We appreciate your suggestion for experiments with Transformers. Unfortunately, training on a large scale task such as WMT translation in a limited time requires a lot of computational resources that we do not have. We would like to note that FFNs in Transformers are small and simple nets with no major impact on the main idea of these networks.
>
> Question 1. We apologize for not reporting the embedding size, d, in the initial version. We have fixed this issue in the revised version.

---

### Decision · Program_Chairs · 2019-12-19

**Decision:**

Reject

**Comment:**

This paper proposes a neural network architecture that represents each neuron with input and output embeddings. Experiments on CIFAR show that the proposed method outperforms baseline models with a fully connected layer.

I like the main idea of the paper. However, I agree with R1 and R2 that experiments presented in the paper are not enough to convince readers of the benefit of the proposed method. In particular, I would like to see a more comprehensive set of results across a suite of datasets. It would be even better, although not necessary, if the authors apply this method on top of different base architectures in multiple domains. At the very least, the authors should run an experiment to compare the proposed approach with a feed forward network on a simple/toy classification dataset. I understand that these experiments require a lot of computational resources. The authors do not need to reach SotA, but they do need to provide more empirical evidence that the method is useful in practice.

I also would like to see more discussions with regards to the computational cost of the proposed method. How much slower/faster is training/inference compared to a fully connected network?

The writing of the paper can also be improved. There are many a few typos throughout the paper, even in the abstract.

I recommend rejecting this paper for ICLR, but would encourage the authors to polish it and run a few more suggested experiments to strengthen the paper.